# Toxicity and Control Efficacy of an Organosilicone to the Two-Spotted Spider Mite *Tetranychus urticae* and Its Crop Hosts

**DOI:** 10.3390/insects13040341

**Published:** 2022-03-30

**Authors:** Jin-Cui Chen, Zhong-Zheng Ma, Ya-Jun Gong, Li-Jun Cao, Jia-Xu Wang, Shao-Kun Guo, Ary A. Hoffmann, Shu-Jun Wei

**Affiliations:** 1Institute of Plant Protection, Beijing Academy of Agricultural and Forestry Sciences, 9 Shuguanghuayuan Middle Road, Haidian District, Beijing 100097, China; chenjincui1314@126.com (J.-C.C.); m1394829269@163.com (Z.-Z.M.); gongyajun200303@163.com (Y.-J.G.); gmatjhpl@163.com (L.-J.C.); jxwang1999@163.com (J.-X.W.); mscgsk@163.com (S.-K.G.); 2School of BioSciences, Bio21 Institute, The University of Melbourne, Parkville 3052, Australia; ary@unimelb.edu.au

**Keywords:** *Tetranychus urticae*, organosilicone, trisiloxane ethoxylate, acaricide, crop safety

## Abstract

**Simple Summary:**

The organosilicone is commonly used as surfactant ingredient in agriculture. Some pilot studies showed pesticidal activities of some organosilicone surfactants. This study examined the toxicity of an organosilicone, Silwet 408, to the two-spotted spider mite *Tetranychus*
*urticae* and explored the field usage of this chemical in pest control. We tested this chemical on eggs, nymphs, and adults of *T. urticae*. We applied a two-sprays strategy to enhance the field control efficacy based on the high toxicity of Silwet 408 to nymphs and adults and the lack of toxicity to eggs of *T. urticae*. However, their phytotoxicity should be taken under consideration. Our study improved our understand of the toxicity and safety of organosilicone surfactants. The results provide a basis for the usage of this chemical in mite control.

**Abstract:**

Organosilicone molecules represent important components of surfactants added to pesticides to improve pest control efficiency, but these molecules also have pesticidal properties in their own right. Here, we examined toxicity and control efficacy of Silwet 408, a trisiloxane ethoxylate-based surfactant, to the two-spotted spider mite (TSSM), *Tetranychus urticae* and its crop hosts. Silwet 408 was toxic to nymphs and adults of TSSM but did not affect eggs. Field trials showed that the control efficacy of 1000 mg/L Silwet 408 aqueous solution reached 96% one day after spraying but declined to 54% 14 days after spraying, comparable to 100 mg/L cyetpyrafen, a novel acaricide. A second spraying of 1000 mg/L Silwet 408 maintained control efficacy at 97% when measured 14 days after spraying. However, Silwet 408 was phytotoxic to eggplant, kidney bean, cucumber, and strawberry plants, although phytotoxicity to strawberry plants was relatively low and declined further seven days after application. Our study showed that while the organosilicone surfactant Silwet 408 could be used to control the TSSM, its phytotoxicity to crops should be considered.

## 1. Introduction

The two-spotted spider mite (TSSM), *Tetranychus urticae* Koch, is an important arthropod pest that feeds on more than a thousand plant species, including vegetables, fruit trees, cotton, and corn [1,2]. Due to its small body size, high fecundity, and extremely high pesticide resistance, TSSM is one of the most difficult pests to control in agricultural production systems [3,4,5,6]. Many methods have been developed to control TSSM, such as biological control using predatory mites [7,8,9], chemical control using various acaricides [10,11], and plant-based control using resistant cultivars and intercropping [12,13].

Chemical control is the most commonly-used method for managing TSSM due to its easy application and low economic cost [14,15,16]. However, TSSM has developed resistance to almost all chemical classes of acaricides applied against it. TSSM shows resistance even to those acaricides that have only been in use for a few years [4,17]. Resistance in TSSM has been reported in more than 40 countries [18,19,20,21].

Spray adjuvants are often used in applications of pesticides to increase control efficacy [22,23,24]. Adjuvants can substantially reduce the usage of active ingredients required to control various pests and diseases [25]. Organosilicone surfactants, especially trisiloxane surfactants, have been applied as efficient adjuvants with herbicides, desiccants, defoliants, insecticides, acaricides, fungicides, plant growth regulators, and foliar nutrients [26,27,28,29]. They have excellent wettability and spreadability to reduce the surface tension of water, thereby making the distribution of active ingredients on waxy or hydrophobic surfaces more uniform [30].

Some studies have found that organosilicone products show pesticidal activity to several pests, such as spider mites, aphids, citrus leaf miners, and armyworms [31,32,33,34]. The organosilicone surfactants may achieve effective control against pests through mechanical respiratory inhibition or interference with critical physiological processes [12,32,35], although this depends on the nature of the chemicals, target pests, and developmental stages of the pests. An organosilicone surfactant (Silwet L-77) showed high lethal bioactivity against the immature/adult stages of the melon aphid *Aphis gossypii*, western flower thrips *Frankliniella occidentalis*, and the Pacific spider mite *Tetranychus pacificus* [26,36], nymphs of the silverleaf whitefly *Bemisia argentifolii* [37], and nymphs but not eggs and adults of the Asian citrus psyllid *Diaphorina citri* [25]. Cowles, Cowles, Mcdermott, and Ramoutar [32] found that three organosilicone surfactants (Silwet L-77, Silwet 408, and Silwet 806) were toxic to adult TSSM, although other life stages were not considered even though effective control will depend on effects across life stages given that this pest has overlapping generations.

Despite organosilicone surfactants holding promise in pest control, they also have potential environmental risks [38]. The organic silicone adjuvant, Breakthru S240, may have impacted the growth, breeding, and sex differentiation of a small planktonic crustacean *Daphnia magna* (Crustacea: Phyllopoda) at 0.2 mg/L [39]. Several organosilicone surfactant adjuvants (Dyne-Amic, Syl-Tac, Sylgard 309, and a modified trisiloxane) were detected harming honey bee olfactory learning more than other nonionic adjuvants (Activator 90, R-11, and Induce) [40]. While Silwet L-77 showed no phytotoxicity or negative effects on shoot growth of citrus [25] and table grape [26], it caused severe phytotoxicity in tomato leaves [37].

In this study, we investigated the toxicity and field control efficacy of the organosilicone surfactant, Silwet 408, to TSSM. We further evaluated the safety of this chemical to crops when applied to common host plants of TSSM (strawberry, eggplant, cucumber, and kidney bean).

## 2. Materials and Methods

### 2.1. Testing Chemicals

The trisiloxane ethoxylate-based surfactant, Silwet 408, was produced by General Electric Company (Boston, MA, USA). We chose cyetpyrafen, a complex II inhibitor commonly used against spider mites, as a positive control in field trials [41,42]. The 30% cyetpyrafen suspoemulsion (SE) was produced by Shenyang Sciencreat Chemicals Co., Ltd., (Shenyang, China).

### 2.2. Testing Mites

A laboratory population of TSSM was used in toxicity tests. This population was collected from strawberries in Xiaoshan District, Hangzhou City, Zhejiang Province, and had been reared on kidney bean (Phaseolus vulgaris) in the laboratory at 25 °C, 60 ± 5% relative humidity, and a 16:8 L:D photoperiod without exposure to acaricides for one year prior to experiments.

### 2.3. Bioassay

For the bioassay, we used 100 mL transparent plastic cups (4.5 cm in height, 5.5 cm in bottom diameter, 6.5 cm in top diameter) to keep the TSSMs. A layer of 0.2% agar was placed on the bottom of the cups to avoid leaves drying out. Fresh leaves of kidney bean were cut to fit the cup and put onto the agar with the lower side on top. The borders of the leaves were sealed using 0.2% agar. The Silwet 408 was diluted into multiple concentrations from 167 to 1000 mg/L with distilled water. About twenty nymphs or female adults of TSSM were transferred to the prepared kidney bean leaf in the cup. Four biological replicates were conducted for each treatment and control. Individual cups with TSSMs were immediately sprayed with a 5 mL solution of Silwet 408 in the upper vial of a Potter Spray Tower (Burkard Scientific, London, UK) at 68.9 kPa. Then, we covered the cup containing treated TSSMs with a layer of tissue paper to prevent escape of the TSSMs and to absorb moisture. The tissue paper was fixed using a plastic cup cover with a 2 cm diameter hole. Mortality was assessed under a Stemi 305 stereomicroscope (Zeiss, Jena, Germany) 24 h or 48 h after spraying. The TSSMs were considered dead if no movement of appendages was observed when they were prodded with a fine brush.

The toxicity of Silwet 408 to eggs of TSSM was determined following a standard method for mites recommended by the FAO (Food and Agriculture Organization of the United Nations) [43,44]. Five mated female adults were allowed to oviposit on a kidney bean leaf disc of 3 cm diameter for 24 h. About 20 eggs were moved to a double-sided tape on a glass slide. Three concentrations (1000, 333, and 200 mg/L) of Silwet 408 aqueous solution were used to treat the eggs of TSSM. The slides were dipped in a test solution for 5 s. The number of hatched eggs was counted daily. Four biological replicates were conducted for each treatment and control. The treated nymphs, adults, and eggs of TSSM were kept at 25 °C, 60 ± 5% relative humidity, and a 16:8 L:D photoperiod. We used distilled water as a control. Four biological replicates were conducted for each treatment and control of bioassays for eggs.

The corrected mortality was calculated as:

Corrected mortality = (mortality in treatment group − mortality in control group)/(1 − mortality in control group).

### 2.4. Effect of Storage on the Acaricidal Activity of Silwet 408 Aqueous Solutions

To examine storage time on the acaricidal activity of Silwet 408 aqueous solutions, we stored the 1000 mg/L solution, a recommended concentration for field control of TSSM, for 4, 8, 16, 24, and 48 h. Then, the stored solutions were used for bioassays with TSSM female adults, as described above.

### 2.5. Field Trial on Control Efficacy of Silwet 408

The control efficacy of Silwet 408 to TSSM was tested on strawberry plants grown in a greenhouse in Beijing, China. The greenhouse was 60 m long, 8 m wide, and 4.3 m high. Strawberries of the “beauty” cultivar were planted on 28 August 2019. The TSSMs occurred naturally on the strawberry plants. The experiments were conducted in March 2020 when the strawberry plants were in the middle fruiting stage. Before spraying treatments, a compound leaf was marked and the number of nymph and adult TSSMs on the leaves was counted. Three concentrations of Silwet 408 diluted with water were sprayed evenly on the back and top sides of leaves (333, 500, and 1000 mg/L). We used the acaricide cyetpyrafen as a positive control and water as a negative control. Cyetpyrafen was diluted with distilled water to obtain a concentration of 100 mg/L. Spraying was carried out using a 15-LTC Electric Sprayer (Matabi, Goizpers C.L., Antzuola, Spain) with an application of 900 kg solution per hm^2^. Eight replicates were conducted for each treatment using ten strawberry plants as a replicate. The number of nymphs and adults of TSSM was counted and recorded 1, 3, 7, and 14 days after the treatments.

To cope with a declining control efficacy, we sprayed three concentrations of Silwet 408 again on previously treated strawberries 14 days after the first treatment. Four replicates were conducted for each treatment using ten strawberry plants as a replicate. The number of nymphs and adults TSSM was counted and recorded 14 days after the second spraying.

The dropping rate of TSSMs was calculated as:

Dropping rate of TSSM (%) = 100 × (number of TSSMs before spraying − number of TSSMs after spraying)/number of TSSMs before spraying. This measure can be negative if there is an increase in mite numbers on leaves across count times, which happened in some controls (see Results). Therefore, we also calculated the control efficacy of the treatments as:

Control efficacy (%) = 100 × (dropping rate of TSSMs of the treatment − dropping rate of TSSMs of the control)/(100 − dropping rate of TSSM of control).

### 2.6. Impact of Organosilicon on Different Crops

Four host plants of TSSM (strawberry (beauty), eggplant (Jingqie No.6), kidney bean (Yunpin No. 1), and cucumber (Jinchun No. 3)) were used to estimate the potential adverse effects of Silwet 408 on crops. The tested strawberry, eggplant, kidney bean, and cucumber plants had 5–7 compound leaves, 3–4 leaves, 2–3 leaves, or 5–6 leaves, respectively. Crops were treated with 1000, 2000, or 4000 mg/L Silwet 408. The spraying treatments were conducted using a 500 mL volume manual sprayer. The application rates of Silwet 408 on strawberry, eggplant, kidney bean, and cucumber plants were 18, 9, 13, and 8 mL/crop, respectively. The treated crops were transferred to incubators set at 20, 25, 30, or 35 °C and with a 16:8 L:D photoperiod. Each treatment had three replicates. The phytotoxicity of Silwet 408 to these crops was investigated 1, 3, and 7 days after treatment according to the Pesticide-Guidelines for Field Efficacy Trials (GB/T 17980.28-2000) required by the Ministry of Agriculture of China. The classification criteria in these guidelines are as follows:Grade 0: no injury;Grade 1: the injury area accounts for more than 5% of the whole leaf area;Grade 3: the injury area accounts for more than 6–10% of the whole leaf area;Grade 5: the injury area accounts for more than 11–25% of the whole leaf area;Grade 7: the injury area accounts for more than 26–50% of the whole leaf area;Grade 9: the injury area accounts for more than 50% of the entire leaf area.

The phytotoxicity index of treated plants was calculated as:

Phytotoxicity index = Σ [(number of injured leaves at grade i × corresponding grade i)/(total leaf number × 9)] × 100.

### 2.7. Data Analysis

The TSSM (egg, nymph, and female adult) mortality and corrected mortality was calculated using Abbott’s formula [45]. Six doses of Silwet 408 were tested to calculate LC_50_. The lethal concentration of 50% (LC_50_) of treated TSSMs (nymph and female adult) and its 95% confidence intervals was calculated based on mortality by using a probit analysis implemented in DPS v12.01 (DPS software, Hangzhou, China). The difference between the LC_50_ values was determined by the ratio of lethal concentrations; if the 95% confidence intervals of the ratio between two concentrations crosses 1 (e.g., 95% CI = 0.9–1.1), there is no difference between two lethal concentrations [46]. Statistical analyses on corrected mortality, control efficacy, and the phytotoxicity index were conducted in SPSS 20.0 (SPSS Inc., New York, NY, USA). The normal distribution and variance homogeneity were measured, and then ANOVAs were applied. One-way ANOVAs followed by Tukey HSD tests were undertaken to test statistical differences in dropping rate and control efficacy among treatments. A multi-way ANOVA with fixed factors was first conducted to determine the effects of concentration, days after treatment, crop species, temperature, and their interactions on the phytotoxicity index. This was followed by three-way ANOVAs for each crop after treatment to investigate the effects of concentration, temperature, time, and interactions on the phytotoxicity index. Treatment values are shown as mean values and standard errors.

## 3. Results

### 3.1. Toxicity of Silwet 408 to TSSM

Bioassay results showed that Silwet 408 has high toxicity to adults and nymphs of TSSM with an LC_50_ of 291 and 427 mg/L, respectively, 24 h after spraying, and 282 and 386 mg/L respectively, 48 h after spraying (Table 1). There was some overlap of 95% confidential limits of LC_50_ values for the adult and nymph stages when examined 24 h after spraying but not at 48 h. These results suggest similar toxicity of Silwet 408 to adults and nymphs. However, Silwet 408 lower than 200 mg/mL indicated a lack of toxicity against adult TSSM, which makes the regression not significant. The corrected mortality of TSSM eggs treated with 200, 333, and 1000 mg/L Silwet 408 aqueous solution was −1.66 ± 4.58%, 4.00 ± 4.71%, and −1.84 ± 2.16% respectively, when examined 48 h after spraying (Table 1, Appendix A). These results indicate a lack of toxicity of Silwet 408 when applied to TSSM eggs.

Female adult TSSMs were treated with 1000 mg/L of Silwet 408 aqueous solution stored for 4, 8, 16, 24, and 48 h at room temperature after dilution. The treated adults all died 24 h after spraying treatments, indicating that 48 h storage after dilution did not affect the acaricidal activity of Silwet 408.

### 3.2. Control Efficacy of Silwet 408 to TSSMs in Field

One day after the first spraying, the dropping rate of the treated TSSMs was 96% and 97% for 1000 and 500 mg/L Silwet 408, respectively, and 94% for 100 mg/L cyetpyrafen (Table 2). The dropping rate of mites treated with 100 mg/L cyetpyrafen and 1000 and 500 mg/L Silwet 408 was significantly higher than that of 333 mg/L Silwet 408 and water treatments (*F* = 227.885, *df* = 4, 35, *p* < 0.001) (Table 2). When we examined the dropping rate on day fourteen, it declined to 28%, −24%, −32%, and 40%, for 1000, 500, and 333 mg/L Silwet 408 and 100 mg/L cyetpyrafen treatments, respectively. On day fourteen after spraying application, TSSMs treated with 1000 mg/L Silwet 408 or 100 mg/L cyetpyrafen showed a higher dropping rate than 500 mg/L Silwet 408, 100 mg/L Silwet 408, and water treatments (F = 13.205, df = 4, 15, *p* = 0.008) (Table 2).

The control efficacy differed among treatments one day after spraying treatment (*F* = 21.582, *df* = 3, 28, *p* = 0.006); the efficiency of 1000 mg/L Silwet 408 was similar to that of 100 mg/L cyetpyrafen or 500 mg/L Silwet 408. The control efficacy of 1000 mg/L Silwet-408 was approximately equivalent to that of 100 mg/L cyetpyrafen, and their control efficacy was significantly higher than that of 500 and 333 mg/L Silwet 408 treatments (*F* = 35.867, *df* = 3, 12, *p* = 0.006); Silwet 408 with concentrations of 333 and 500 mg/L showed the lowest control efficacy compared with the other treatments, with values of 16% and 21%, respectively (Table 2).

We applied another spray treatment 14 days after the first spraying on the same strawberries; the dropping rate was 95%, 80%, 71%, and 76% for 1000, 500, and 333 mg/L Silwet 408 and 100 mg/L cyetpyrafen, respectively, on day 14 after the second spraying and treatments differed significantly (*F* = 10.177, *df* = 3, 12, *p* = 0.046). The control efficacy of 1000 mg/L of Silwet 408 was 97%, significantly higher than for the other treatments (Table 2). These results indicate that a second spraying application of Silwet 408 increased the control efficacy of TSSMs compared to a single application.

### 3.3. Phytotoxicity of Silwet 408 to Crops

Silwet 408 at concentrations of 4000, 2000, and 1000 mg/L led to various degrees of phytotoxicity symptoms on strawberries, eggplants, kidney beans, and cucumbers (Appendix A, Figure 1). The ANOVA indicated a significant difference in the interactions between concentration, temperature, and crop as well as between concentration, time, and crop. There was a tendency for more damage to occur at higher concentrations, but for strawberries, this effect was weaker at 7 days, and for eggplants, this effect was weaker after one day (Figure 2). For kidney beans, damage was substantial at 20 °C when the lower concentration was tested, whereas temperature effects were relatively minor for other crops such as eggplants. Time and crop showed a strong two-way interaction (Table 3), reflecting the absence of much effect of time on damage to cucumber and eggplants, but stronger effects on strawberry, where a decline was observed, and kidney bean, where damage increased with time (Figure 2). These results indicated that different crops reacted in diverse ways to Silwet 408 and damage to strawberry was particularly low after 7 days regardless of the concentration applied and temperature tested. On day one, the phytotoxicity index of strawberries treated using 4000 mg/L Silwet 408 ranged from 36 to 66 at different temperatures, which was significantly lower than that of other treated crops (64–71 for eggplant, 46–72 for kidney bean, 67–74 for cucumber) (*F* = 31.871–401.814, *df* = 3, 8, *p* < 0.015). For treatment of 1000 mg/L Silwet 408, the phytotoxicity index of eggplant was higher (12–17) than strawberry (5–10), cucumber (0.6–3), and kidney bean (3–9) (Appendix A).

## 4. Discussion

Organosilicone surfactants are commonly used as synergists of pesticides because of their favorable wettability, spreadability, adhesivity, and penetrability [22,23,24]. Our study investigated the acaricidal activity of an organosilicone surfactant, Silwet 408, and evaluated its effect on controlling TSSM in greenhouse strawberry plants. In addition, we tested the phytotoxicity of Silwet 408 to four common host plants of TSSM.

A previous study had shown that the trisiloxane surfactant, Silwet 408, was toxic to adults of TSSM [32]. In this study, we found that Silwet 408 was also toxic to TSSM nymphs. However, Cowles, Cowles, Mcdermott, and Ramoutar [32] found that the LC_50_ of Silwet 408 was 5.46 mg/L, much lower than the 282 mg/L value obtained for adults in our study. In the previous study, the toxicity of Silwet 408 was tested using the dipping method, which led the tested TSSMs being immersed in the surfactant solutions. Our study, however, involved contact toxicity, which was achieved by using a spray tower. The leaf dipping method exaggerates the degree of interaction between mites and the compound in an environment compared with most foliar application methods [32,36].

To further investigate the control efficacy of Silwet 408 against TSSM in the field, a field trial was conducted with the application of three concentrations of Silwet 408. We found that 1000 mg/L Silwet 408 was comparable with 100 mg/L cyetpyrafen in its effect on TSSM control. On day 14 and after second applications, the control efficacy of Silwet 408 was even higher than that of cyetpyrafen treatments. Although a relatively higher concentration was applied to control TSSM, the organosilicone surfactant physically kills TSSMs, which makes the development of resistance unlikely.

Although Silwet 408 was effective against TSSM, it had minor or no effect on TSSM eggs. This organosilicone surfactant may act through respiratory inhibition by permitting water to infiltrate the trachea or peritremes of pests [47,48]. It is, therefore, to be expected that TSSM eggs are more tolerant to the effects of Silwet 408. We applied a second spraying of Silwet 408 to tackle this issue given that hatched eggs would be exposed after the first spraying. Our field trial demonstrated a 97% control efficacy 14 days after the second spraying. In addition, the combined use of other acaricides to kill eggs provides a different option to ensure control. Toxicity can, however, vary among different binary mixtures of Silwet 408 and other acaricides, and more studies should be conducted to evaluate whether there is synergism, additivity, or antagonism of Silwet 408 with selected chemicals [49].

Organosilicone surfactants have shown promising potential in TSSM management, and the other risks associated with these adjuvants should be evaluated. To this end, we conducted a phytotoxicity test on four common host plants of TSSM. Silwet 408 showed phytotoxicity to all these host plants. Disease spots appeared on the leaves of strawberry, cucumber, eggplant, and kidney bean. The phytotoxicity index of four crops indicated that Silwet 408 exhibited strong phytotoxicity on eggplant and kidney bean, but relatively low phytotoxicity on strawberry and cucumber. The phytotoxicity index of strawberry plants declined on day seven after treatment, indicating that the symptoms of phytotoxicity recovered through time. Previous studies have also suggested that the phytotoxicity of organosilicone adjuvants varies among plants [25,26,40]. Our study showed that Silwet 408 could be used to control TSSMs on strawberries. However, the potentially toxic effects of Silwet 408 on other non-target organisms, such as honey bees and natural enemies, need further investigation. Relative humidity of the environment could also affect the effectiveness of trisiloxane surfactants, given that Silwet L-77 was particularly effective against the green peach aphid *Myzus persicae* under high humidity conditions [32,36].

## 5. Conclusions

Our results demonstrate the possibility of using Silwet 408 as a novel acaricide for TSSM management. However, the practicality of using Silwet 408 under field conditions, particularly where pollinators and predators are present, still needs further investigation.

## Figures and Tables

**Figure 1 insects-13-00341-f001:**
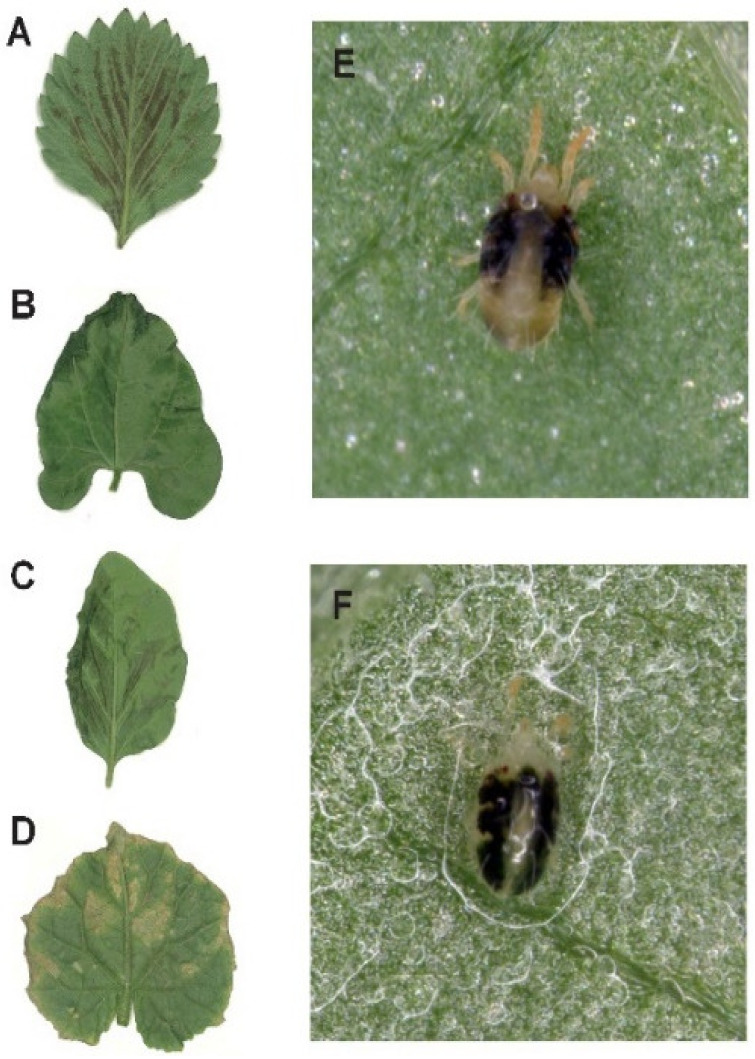
Phytotoxicity symptoms of treated crops and status of treated *Tetranychus urticae* adults by water and Silwet 408. (**A**–**D**) Phytotoxicity symptoms of strawberry, kidney bean, eggplant, and cucumber, respectively. (**E**) Adult of *T. urticae* treated using water. (**F**) Adult of *T. urticae* treated using Silwet 408 aqueous solution.

**Figure 2 insects-13-00341-f002:**
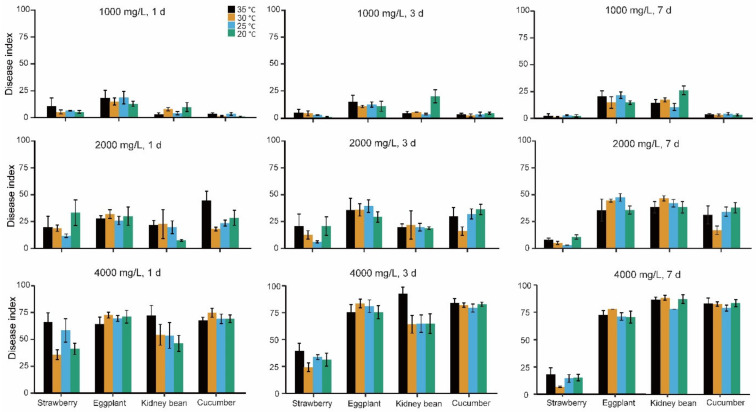
Effect of temperature on phytotoxicity of Silwet 408 to four crops. Plants were treated with 1000, 2000, and 4000 mg/L Silwet 408, respectively. The phytotoxicity of Silwet 408 to these plants was investigated 1, 3, and 7 d after applications. The descriptive statistics are shown as the mean value and standard errors of the mean.

**Table 1 insects-13-00341-t001:** Toxicity of the organosilicone Silwet 408 to different developmental stages of *Tetranychus urticae*.

Stage	*n*	Time (Hour)	Regression Equation	SE of Slope	LC_50_ (95% CL) (mg/L)	*df*	*χ* ^2^
Adult	20	24	11.42x − 23.14	1.27	291.07 (249.13–418.39)	3	34.87
		48	17.36x − 37.53	2.37	281.60 (266.40–307.17)	2	1.93
Nymph	20	24	6.76x − 12.79	0.65	427.05 (397.31–467.50)	4	13.26
		48	5.67x − 9.67	0.59	386.52 (321.23–585.00)	3	17.16

*n*, sample size per dose-group; LC_50_, lethal concentration 50; CL, 95% confidential limit; ***χ*^2^**, chi-square.

**Table 2 insects-13-00341-t002:** Control efficacy of the organosilicone Silwet 408 to *Tetranychus urticae* on greenhouse strawberry as measured by dropping rate (DR) and control efficiency (CE).

Chemical	Con. (mg/L)	Day 1		Day 3		Day 7		Day 14		Day 14 after 2nd Spray	
		DR (%)	CE (%)	DR (%)	CE (%)	DR (%)	CE (%)	DR (%)	CE (%)	DR (%)	CE (%)
Silwet 408	1000	95.8 ± 1.6a	96.1 ± 1.6a	89.3 ± 2.4a	91.8 ± 2.0a	53.5 ± 4.6ab	69.8 ± 3.1a	28.4 ± 5.8a	54.2 ± 2.0a	94.9 ± 1.5a	97.2 ± 0.9a
	500	96.7 ± 0.7a	97.0 ± 0.6a	88.4 ± 2.4a	91.0 ± 1.8a	55.9 ± 4.9ab	72.1 ± 2.9a	−24.0 ± 12.4b	20.7 ± 5.2b	79.7 ± 6.0a	89.5 ± 2.3b
	333	65.3 ± 5.7b	68.6 ± 5.1b	67.9 ± 4.1ab	75.4 ± 2.8b	31.9 ± 5.3a	56.1 ± 4.7b	−32.3 ± 11.3b	15.8 ± 1.5b	71.2 ± 2.3a	84.7 ±0.5b
Cyetpyrafen	100	93.8 ± 1.3a	94.4 ± 1.2a	83.6 ± 3.4b	87.0 ± 2.8a	81.1 ± 3.7b	88.4 ± 1.9c	40.3 ± 5.5a	61.4 ± 3.4a	76.0 ± 3.0a	87.1 ± 1.5b
Water	-	−10.4 ± 1.8c	-	−30.1 ± 6.5c	-	−62.0 ± 14.4c	-	−56.7 ± 12.0b	-	−89.1 ± 16.9b	-

Means for each treatment in the same column followed by the same lowercase letter are not significantly different (*p* < 0.05, one-way ANOVA with Tukey HSD test). Con., concentration.

**Table 3 insects-13-00341-t003:** Effects of temperature, concentration, examination days after the spray application, and crop on phytotoxicity index of four host plants of *Tetranychus urticae*.

Source	*df*	*F*	*p*
Concentration (con.)	2	973.941	<0.001
Temperature (tem.)	3	2.410	0.067
Time	2	3.462	0.033
Crop	3	107.873	<0.001
Con. × tem.	6	0.901	0.495
Con. × time	4	1.637	0.165
Con. × crop	6	31.464	<0.001
Tem. × time	6	0.629	0.707
Tem. × crop	9	1.597	0.115
Time × crop	6	20.858	<0.001
Con. × tem. × time	12	0.414	0.958
Con. × tem. × crop	18	2.628	<0.001
Con. × time× crop	12	3.718	<0.001
Tem. × time× crop	18	0.450	0.975
Con. × tem. × time× crop	36	0.385	1.000
Error	288		
Total	431		

## Data Availability

Not applicable.

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
