# Peer review of "Toxicity and Control Efficacy of an Organosilicone to the Two-Spotted Spider Mite Tetranychus urticae and Its Crop Hosts"

_insects, 2022, doi:10.3390/insects13040341_

Round 1

Reviewer 1 Report

Reviewer’s comments

to the authors about the manuscript presented to Insects “Toxicity and control efficacy of an organosilicone to the two-spotted spider mite Tetranychus urticae and its crop hosts”

The manuscript presents an interesting study that investigates the potential of an adjuvant to control Tetranychus urticae, together with secondary effects on several crops. For this reason, I consider that the manuscript has merit to be published in the journal, but some questions have been arisen while reviewing it.

Comments to the authors:

Line 98. Indicate that only one cup (with the 20 individuals) were tested in each concentration. Below I suggest to indicate the number of concentrations tested.

Line 99. Authors indicate that the volume used in the application was of 5 ml. This is the volume of solution put in the upper vial, isn’t it?.

In my opinion it would be very informative to know the amount of volume that reached the leaves. Especially if comparations with other studies should be done.

Line 109-110. The number of eggs (or an intervale) should be included.

Line 114. Clarify that this number of replicas were used with the test with eggs.

Lines 131-132. “900 kg/hm2”. Is this correct?. It seems extremely high those 900 kg in 100 m2 . Please revise.

Lines 167-168. I suggest the following correction in the expression:

Phytotoxicity index = Σ [(number of injured leaves at grade i × corresponding

grade i) / (total leaf number × 9)] × 100

Lines 176-177. No transformation was needed.? Normality was fulfilled?. In such case state that in the text. It is rather usual to use the transformation “arc sin √(p)” to analyse data expresed as proportions.

Table 1. Df column varies with 3 and 2, 4 and 3. Why this difference in df column? Did you use different number of concentrations for adults and nymphs,with 24 and 48 hours?

If that is the case you should indicate in Material and Methods the number of concentrations tested (line 97), and not only make a general comment.

Table 1. The value X2=“1.93” with 2 df is not significative at 5 %. If it is not a mistake you must indicate that such regression (Adult, 48 h) was not significative, and so no effect on mortality was observed.

Line 207. Check the values “df = 4, 35, “

5 treatments -1 = 4

5 treatments x 4 replicates-1 = 19 = total df

error= 19-4 =15

Compare with line 214, where df seems correct.

Line 220. Check de values “df = 3, 28”

4 treatments -1 =3

4 treatments x 4 replicates-1 = 15 = total

error= 15-4 =12

Compare with line 222, where df seems correct.

Line 235. “The ANOVA indicated significancy in the three-way…”

Figure 2. The same left scale should be used in all the graphs. It is a bit confusing to see the graphs and not realise of the different scales. Alternatively, authors could warn the reader with this aspect.

Another suggestion: at least put right scale of the three graphs at the left with the same value (40).

Table 3. The column “df” summs a value of Total=432, but it seems that should be higher:

4 species x 3 conc. x 4 temp. x 3 dates x 4 replicas = 576 -1 =575

and the error df accordingly should be higher.

Lines 273-275. The two pesticide application methods were quite different, and their results differs greatly.

In both methods is not clear the quantity of product that could arrive to the mites. If it is possible, such information should be included in the Material and Methods, as indicated previously.

Author Response

  1. Line 98. Indicate that only one cup (with the 20 individuals) were tested in each concentration. Below I suggest to indicate the number of concentrations tested.

Response: We used four cups (replicates). It was added (See L104-105).

  1. Line 99. Authors indicate that the volume used in the application was of 5 ml. This is the volume of solution put in the upper vial, isn’t it?.

Response: Yes, 5 ml solution of silwet 408 was put in the upper vial. We have added the information (See L106).

  1. Line 109-110. The number of eggs (or an intervale) should be included.

Response: Added (See L116-117).

  1. Line 114. Clarify that this number of replicas were used with the test with eggs.

Response: We have added the information of replications used with eggs (See L120-121).

  1. Lines 131-132. “900 kg/hm2”. Is this correct?. It seems extremely high those 900 kg in 100 m2 . Please revise.

Response: Yes, it should be 900 kg solution per hm2. Revised (See L144).

  1. Lines 167-168. I suggest the following correction in the expression:

Phytotoxicity index = Σ [(number of injured leaves at grade i × corresponding

grade i) / (total leaf number × 9)] × 100

Response: Done (See L182-183).

  1. Lines 176-177. No transformation was needed.? Normality was fulfilled?. In such case state that in the text. It is rather usual to use the transformation “arc sin √(p)” to analyse data expresed as proportions.

Response: Yes, the data fulfill the normality.

  1. Table 1. Df column varies with 3 and 2, 4 and 3. Why this difference in df column? Did you use different number of concentrations for adults and nymphs, with 24 and 48 hours?

Response: We set the same number of concentrations. However, in some high concentrations, all individuals died; thus, some concentrations were excluded from analysis.

  1. Table 1. The value X2=“1.93” with 2 df is not significative at 5 %. If it is not a mistake you must indicate that such regression (Adult, 48 h) was not significative, and so no effect on mortality was observed.

Response: Thanks for your comments. The regression was not significant due to no considerable mortality effect on treated adults with the application of Silwet 408 lower than 200 mg/ml. We have added relevant descriptions in the manuscript (See L211-212).

  1. Line 207. Check the values “df = 4, 35, “

Response: Thanks for your suggestion. We wrote the wrong number of replicates. Eight replicates were conducted for the first spray treatment. In that case, the df value was correct (See L144-145).

  1. Line 220. Check de values “df = 3, 28”

Response: We have corrected the number of replicates prepared for the first spray application. Eight replicates were conducted for the first treatment, while four replicates were prepared for the second spray treatment (See L142-145, 149-150).

  1. Line 235. “The ANOVA indicated significancy in the three-way…”

Response: We have revised the sentence (See L266-267).

  1. Figure 2. The same left scale should be used in all the graphs. It is a bit confusing to see the graphs and not realise of the different scales. Alternatively, authors could warn the reader with this aspect. Another suggestion: at least put right scale of the three graphs at the left with the same value (40).

Response: We have revised figure 2.

Table 3. The column “df” summs a value of Total=432, but it seems that should be higher:

4 species x 3 conc. x 4 temp. x 3 dates x 4 replicas = 576 -1 =575 and the error df accordingly should be higher.

Response: We wrote the wrong number of replicates. Three replicates were prepared for determining the safety of Silwet 408. The df value of the total is 431, and the error df is still 288 (See L170 and table 3).

Lines 273-275. The two pesticide application methods were quite different, and their results differs greatly. In both methods is not clear the quantity of product that could arrive to the mites. If it is possible, such information should be included in the Material and Methods, as indicated previously.

Response: The quantity of Silwet 408 that exactly arrived on mites can not be counted. Five ml of Silwet 408 was put into a vial upon the spray tower for testing the toxicity of Silwet 408 to TSSMs..

Reviewer 2 Report

Authors explore the effects of the organosilicone surfactant Silwet 408 spray application on the mortality of different stages of spider mite and the phytotoxicity of this compound on spider mite host crops. The results of this study show pros and cons of the use of Silwet 408 as an acaricide in the field and might be helpful in the research of new compounds used for pest control.

In my opinion, the manuscript is easy to read, language is correct, the sections are well structured and the experiments are coherent and well designed. 

However, I found some issues that should be addressed prior to publication:

Line 22. Its crop hosts

Line 81. I suggest to split testing chemicals and mites in two materials and methods sections.

Line 170. Please indicate in materials and methods how many doses/the range of doses tested to calculate LC50.

Line 173. Authors state that a lack of overlap between 95% confidence intervals was taken as a significant difference between LC50 values. That is not correct, a lack of 95% confidence intervals does not mean statistically significant differences. To detect significant differences a statistical test is needed. 

Line 192. Add to materials and methods how corrected mortality is calculated. Is LC50 calculated from corrected mortality?

I was wondering what is TSSM water control mortality in the bioassay, since TSSM are heavily affected by high humidity. 

Line 200. Why do you choose 1000mg/L concentration of Silwet 408 to check its effect on TSSM after 48h storage and not for example, LC50? 1000mg/L concentration is quite high and if some effectivity is lost after 48h and persistence decreases it might not be noticed. 

Line 207. Application of different concentrations of silwet are also different treatments, so this sentence is confusing: “Treatment differences were significant, and dropping rates were significantly higher for these rates than for treatments of 333 mg/L Silwet 408 or water”.

Line 209. “When we examined the dropping rate on days three, seven and fourteen, it had declined to 28%, -24%, -32%, and 40%, for 1000, 500, and 333 mg/L Silwet 408” Droping rates indicated in this sentence correspond only to day 14.

Line 211. “On day fourteen after spray application, TSSMs treated with 1000 mg/L Silwet 408 or 100 mg/L cyetpyrafen showed the highest dropping rate compared with other treatments (Table 2) with treatment differences again significant”.  The statement is confusing, as dropping rates on day 14 are the lowest among the other days tested. In addition, cyetpyrafen DR and CE are the highest.

Line 221. Do you mean: On day 14 after spraying, control efficacy among treatments differed significantly…?

Line 236. “Date” is not accurate, do you mean “time” or days after application?

Line 254. Figure caption of figure 1 A-D. Indicate that leaves are treated with silwet 408 (not water).

Line 311. Please add to conclusions that you found a degree of phytotoxicity to some of T. urticae plant hosts.

Author Response

Line 22. Its crop hosts

Response: Revised (See L22).

Line 81. I suggest to split testing chemicals and mites in two materials and methods sections.

Response: We have split testing chemicals and mites into two sections (See Line92-96).

Line 170. Please indicate in materials and methods how many doses/the range of doses tested to calculate LC50.

Response: Six doses of Silwet 408 were tested to calculate LC50, including 1000 mg/L, 500 mg/L, 333 mg/L, 250 mg/L, 200 mg/L, and 166 mg/L. We have added information in the Materials and method section (See Line188).

Line 173. Authors state that a lack of overlap between 95% confidence intervals was taken as a significant difference between LC50 values. That is not correct, a lack of 95% confidence intervals does not mean statistically significant differences. To detect significant differences a statistical test is needed.

Response: The difference between the LC50 values was determined by the ratio of lethal concentrations; there is no significant difference between two lethal concentrations when the 95% confidence intervals of the ratio between two concentrations contains 1 (See L189-192).

Line 192. Add to materials and methods how corrected mortality is calculated. Is LC50 calculated from corrected mortality?

Response: We have added the calculation formula of corrected mortality. The LC50 was calculated from mortality (See L189-192, L125-127).

I was wondering what is TSSM water control mortality in the bioassay, since TSSM are heavily affected by high humidity.

Response: The mortality of TSSMs treated with water was no more than 10%. After the spray treatments, we covered the cup containing treated TSSMs with a layer of tissue paper to absorb moisture. In that case, humidity could not produce a visible effect on the survival of mites.

Line 200. Why do you choose 1000mg/L concentration of Silwet 408 to check its effect on TSSM after 48h storage and not for example, LC50? 1000mg/L concentration is quite high and if some effectivity is lost after 48h and persistence decreases it might not be noticed.

Response: In practice, at least 1000 mg/L should be used to get a good control efficacy. We would like to check if the 1000mg/L solution will decrease its control efficacy after 48 hours. We also agree with your idea that LC50 is a good choice.

Line 207. Application of different concentrations of silwet are also different treatments, so this sentence is confusing: “Treatment differences were significant, and dropping rates were significantly higher for these rates than for treatments of 333 mg/L Silwet 408 or water”.

Response: We have revised the sentences (See L228-230).

Line 209. “When we examined the dropping rate on days three, seven and fourteen, it had declined to 28%, -24%, -32%, and 40%, for 1000, 500, and 333 mg/L Silwet 408” Droping rates indicated in this sentence correspond only to day 14.

Response: We have revised the sentence (See L233).

Line 211. “On day fourteen after spray application, TSSMs treated with 1000 mg/L Silwet 408 or 100 mg/L cyetpyrafen showed the highest dropping rate compared with other treatments (Table 2) with treatment differences again significant”.  The statement is confusing, as dropping rates on day 14 are the lowest among the other days tested. In addition, cyetpyrafen DR and CE are the highest.

Response: We have revised the sentence (See L232-237).

Line 221. Do you mean: On day 14 after spraying, control efficacy among treatments differed significantly…?

Response: We have revised the sentence (See L249-251).

Line 236. “Date” is not accurate, do you mean “time” or days after application?

Response: We have changed the word “date” to “time” throughout the manuscript.

Line 254. Figure caption of figure 1 A-D. Indicate that leaves are treated with silwet 408 (not water).

Response: We have supplemented the information (See L285-286).

Line 311. Please add to conclusions that you found a degree of phytotoxicity to some of T. urticae plant hosts.

Response: We have supplemented the information (See L331-333).

Round 2

Reviewer 1 Report

Authors have adressed all my questions adequately, and the manuscript has improved.

Reviewer 2 Report

Requested issues have been adressed